# The Impact of Personality Traits on Patient Satisfaction after Telerehabilitation: A Comparative Study of Remote and Face-to-Face Musculoskeletal Rehabilitation during COVID-19 Lockdown

**DOI:** 10.3390/ijerph20065019

**Published:** 2023-03-12

**Authors:** Błażej Cieślik, Tomasz Kuligowski, Luisa Cacciante, Pawel Kiper

**Affiliations:** 1Healthcare Innovation Technology Lab, IRCCS San Camillo Hospital, 30126 Venezia, Italy; 2Faculty of Physiotherapy, Wroclaw University of Health and Sport Sciences, 51-612 Wroclaw, Poland

**Keywords:** telemedicine, SARS-CoV-2, pandemic, satisfaction, remote therapy, eHealth, telehealth, orthopedic treatment, physiotherapy

## Abstract

This study aimed to evaluate the differences in patient satisfaction between telerehabilitation and traditional face-to-face rehabilitation and to identify the impact of personality traits on patient satisfaction with the remote form of rehabilitation. Eighty participants with musculoskeletal pain were recruited for the study. The telerehabilitation group (*n* = 40) completed a single remote session of rehabilitation, whereas the traditional rehabilitation group (*n* = 40) completed a single face-to-face session. After therapy, each participant was asked to complete a tailored satisfaction survey using Google Forms. The Health Care Satisfaction Questionnaire (HCSQ) and the International Personality Item Pool-Big Five Markers-20 (IPIP-BFM-20) were used as outcome measures. Considering the results of patient satisfaction with healthcare service, there were no statistically significant differences between telerehabilitation and traditional rehabilitation groups in the total HCSQ score and its subscales. For the complete HCSQ, agreeableness, conscientiousness, and extraversion were essential predictor variables, accounting for 51% of the variance in patient satisfaction. In conclusion, there were no differences in patient satisfaction between telerehabilitation and traditional rehabilitation groups. In the telerehabilitation group, higher agreeableness levels and lower conscientiousness and extraversion level could predict patients’ satisfaction with telerehabilitation.

## 1. Introduction

During the COVID-19 pandemic, almost all organizational aspects of healthcare services, including patients and healthcare professionals, have had to reorganize and adapt to the new situation [1]. It has been no different with rehabilitation medicine, orthopedic services, and pain management. The treatment of musculoskeletal disorders is primarily related to specific conditions. It can be either operative or non-operative, such as physiotherapy and rehabilitation. While surgical treatment lacks its direct substitute, a conservative treatment was developed through a remote support method—telerehabilitation (TR)—as a part of telemedicine or telehealth [2]. TR was widely known before COVID-19; however, its adoption in clinical settings for musculoskeletal disorders only gradually occurred [3]. Conversely, TR has been widely used in the field of neurorehabilitation, with the emerging literature underlying its efficacy and feasibility for the treatment of motor [4], cognitive [5], speech-language [6,7], and swallowing impairments [8] in neurological disorders. The effectiveness of TR has also been confirmed for the treatment of motor symptoms in patients with multiple sclerosis [9] and Parkinson’s disease [10], suggesting that TR is a valid and feasible method for delivering healthcare services in very different conditions. After COVID-19 spread, the rapid uptake of all branches of telemedicine was a response to forced social distancing.

Since then, TR has been widely used and is considered to be low cost, easy to access, and of acceptable quality compared to the standard approach in both adults and children [11,12,13]. The telerehabilitation approach has been analyzed to improve access to homecare, providing promising results on health outcomes. According to the literature, it might be equivalent to the traditional face-to-face approach when technical and physical requirements (i.e., stable Internet connection, basic exercise devices, accessories, etc.) are met [5,14]. The growing prevalence and positive effects of additional measuring instruments, electronic devices such as wearable sensors, TR-friendly apps, software, and virtual reality (VR) should also be emphasized [15,16,17]. It should also be stated that regardless of COVID-19, telerehabilitation might be a key factor for people with disabilities, both temporary and permanent. Indeed, treatment conducted in patients’ social, educational, and vocational environments can lead to improved functional outcomes and enhanced family and community integration [18]. Therefore, the possibility for the patient to perform treatment at home has positive effects not only in terms of time and cost related to travel to the rehabilitation centers but also in terms of patients’ quality of life (QoL), which in turn could have positive effects on their functional improvements. Many approaches to TR have been proposed in the literature, but standard procedures have not yet been established [19]. The absence of standardized procedures can result in untapped treatment potential, errors, accidents, and unfavorable outcomes in various fields, highlighting the critical need to develop and implement consistent protocols.

On the contrary, TR has disadvantages when applied to patients who might have cognitive impairments, be skeptical, or fear losing some aspects of human interaction [20]. Another significant limitation is the requirement for patients to have access to technology, which may not be feasible for all individuals, especially those living in remote or underdeveloped areas. Whether the patient had previously undergone a specific treatment might also be considered difficult. Those whose therapy was mainly “hands-on” (manual therapy or chiropractic) may face a barrier regarding the exercise-only procedure. Furthermore, patients who struggle with motivation or require more supervision during their rehabilitation may not benefit as much from telerehabilitation as those who do not face these challenges [21]. Kruse and colleagues [22] suggested a list of barriers to the adoption of telemedicine and divided them into several categories including barriers by country (mainly the United States and Europe), barriers for organizations (especially cost and reimbursement), barriers for patients (primarily age and level of education), and barriers for staff and programmers (technically challenged staff and resistance to change).

Barriers and obstacles could affect patient participation, which is crucial in the telerehabilitation process [23]. Determining these factors as precisely as possible will benefit both healthcare professionals and patients. Consequently, it can be useful for establishing protocols or improving the existing ones. Thus, this study aimed to evaluate differences in patient satisfaction between telerehabilitation and traditional face-to-face rehabilitation and to identify the impact of personality on patient satisfaction from the remote form of rehabilitation. We hypothesized that agreeableness and extraversion would be predictors of patients’ satisfaction with telerehabilitation.

## 2. Materials and Methods

### 2.1. Study Setting and Participants’ Characteristics

This study was conducted in May 2020 at a private musculoskeletal rehabilitation clinic in Poland. On 20 March 2020, according to the regulations of the Polish Ministry of Health, an epidemic law was imposed, and temporary physical isolation precautions were taken to prevent the overloading of healthcare systems. Prior to the lockdown, participants included in this study participated in face-to-face rehabilitation, as ordered by a specialist. The participants were referred for rehabilitation for musculoskeletal pain originating from the lower or upper extremities or spine. When government restrictions were announced, some participants were randomly offered remote treatment; therefore, they should be considered as volunteers. Those who continued to receive traditional rehabilitation constituted the control group. Data acquisition and storage procedures were executed in accordance with the ethical standards for human experimentation and the Declaration of Helsinki 1975, revised in Hong Kong in 1989. All participants provided written informed consent to participate in this study.

The sample size for this study was calculated using G*Power 3.1.9.4 software (Heinrich-Heine-University Düsseldorf, Düsseldorf, Germany) with a priori power analysis for *t*-test [24]. We assumed an effect size of 0.75, with a minimum significance (*α*) of 0.05 and statistical power (1 − *β*) of 0.95. To achieve statistical significance, 80 participants were recruited for this study. The telerehabilitation group (*n* = 40) completed a single session (60 min) of a remote form of rehabilitation (conducted via Zoom). The traditional rehabilitation group (*n* = 40) completed a single session (60 min) of rehabilitation at the clinic.

### 2.2. Telerehabilitation and Face-to-Face Rehabilitation

In order to prevent the therapist’s personality from affecting patient satisfaction, a single physiotherapist conducted all therapy sessions. Both in telerehabilitation and face-to-face rehabilitation, all procedures were meticulously chosen according to the standards of physiotherapy in orthopedics with regards to the specific diagnosis.

To account for expected technical limitations and the need for specificity, the telerehabilitation sessions were primarily designed to promote exercise impact and incorporate various forms of autotherapy such as post-isometric relaxation and stretching. Each patient received live instructions, guidance, and assistance from their physical therapist on how to prepare for and perform each exercise. The techniques utilized in the treatment included post-isometric relaxation, stretching, isometric exercises, core stability exercises, automobility exercises (such as Sustained Natural Apophyseal Glides or Mobilizations with Movement), and self-version of specific functional tests.

During face-to-face rehabilitation, all patients underwent a comprehensive set of techniques including both self-treatment and manual therapy. The examination focused on clinical and functional tests with the highest specificity and sensitivity (i.e., SLUMP test, straight leg raise test, passive lumbar extension test, rocking test, and painful catch test for lower-back pain). As a result, the treatment included tools specific to the particular diagnosis: posteroanterior and anteroposterior joint mobilization (based on the guidelines of the International Federation of Orthopedic Manipulative Physical Therapists, IFOMPT), muscle and tissue massage techniques (deep friction, functional, and others), and hands-on controlled core stability exercises.

### 2.3. Outcome Measures

The questionnaire was compiled using Google Forms. After the therapy, each participant was asked to complete a satisfaction survey. Our questionnaire incorporated three questions to assess the participant’s attentiveness, such as requesting them to “*mark answer b*”. A properly completed questionnaire was determined if all three questions were answered correctly. Otherwise, the survey was excluded from the data analysis.

Patient satisfaction was assessed using the Health Care Satisfaction Questionnaire (HCSQ) [25]. It is a self-administered, standardized questionnaire that is designed to gather patients’ feedback on various aspects of their healthcare experience. The questionnaire has 23 items divided into three domains: relationship with the professional, organization of services, and delivery of services and overall satisfaction with the healthcare services received. Participants rated each item on a 4-point Likert scale. The reliability of the scale (Cronbach’s alpha) was 0.78. The total score is reported as a percentage.

Personality traits were assessed using the International Personality Item Pool-Big Five Markers-20 (IPIP-BFM-20) [26]. It is a shortened version of the 50-item Big Five Markers Questionnaire from Goldberg’s International Personality Item Pool [27]. The questionnaire consists of 20 items and measures five personality traits: extraversion, agreeableness, conscientiousness, emotional stability, and intellect. Respondents rated each statement on a Likert scale ranging from “*strongly disagree*” to “*strongly agree*” to indicate how well it describes them. Extraversion is characterized as being outgoing, assertive, and sociable. Agreeableness reflects an individual’s level of empathy, cooperation, and friendliness. Conscientiousness is related to being organized, responsible, and goal-oriented. Emotional stability is related to neuroticism, anxiety, and moodiness. Intellect reflects a person’s level of imagination, creativity, and willingness to try new things. The reliability of the scales (Cronbach’s alpha) ranged from 0.75 to 0.88. Higher scores indicated higher levels of a given trait.

In addition, patients were asked about their pain intensity (using visual analog scale, VAS), pain localization by using an ad hoc body chart (lower/upper extremity or trunk/spine), satisfaction with the form of therapy received (5-point Likert scale), preference for the type of therapy (favors telerehabilitation, favors traditional rehabilitation, or favors neither of the forms), and satisfaction with the quality of the Internet connection (4-point Likert scale, only the telerehabilitation group).

### 2.4. Data Analysis

Data were analyzed using JASP version 0.16.3 (University of Amsterdam, Amsterdam, The Netherlands). We reported the mean and standard deviation (*SD*) for continuous variables. Descriptive statistics for categorical variables were reported as frequency counts and percentages. Baseline demographic variables were compared between groups using unpaired *t*-tests (for continuous variables) and *χ*^2^ tests (for categorical data). The normality of the data distribution was evaluated using the Shapiro–Wilk test. The satisfaction outcome data showed a non-normal distribution; hence, the between-group comparison was performed using the Mann–Whitney *U* test. Stepwise multiple regression was used to determine which personality variables could predict patient satisfaction from the remote form of therapy. Statistical significance was set at *α* < 0.05.

## 3. Results

Of the 80 participants, 76 (95%) were included in the data analysis. Due to incorrect completion of the questionnaire, the results of four participants (one from the telerehabilitation group and three from the traditional rehabilitation group) were excluded from the analysis. The groups did not differ significantly in terms of their baseline characteristics (Table 1). When assessing satisfaction with the Internet connection in a 5-point Likert scale in the telerehabilitation group, 38% (*n* = 15) of the participants marked the answer “*very satisfied*”, 28% (*n* = 11) marked “*satisfied*”, 21% (*n* = 8) marked “*neutral*”, 8% (*n* = 3) marked “*dissatisfied*”, and 5% (*n* = 2) marked “*very dissatisfied*”.

Considering the results related to patient’s satisfaction with healthcare service, there were no statistically significant differences between telerehabilitation and traditional rehabilitation groups in the total HCSQ score and its subscales (Table 2). However, a significant difference was observed in response to the following question: “*Are you satisfied with the therapy form?*”. Participants in the traditional rehabilitation group were approximately 11% more satisfied with this form of therapy (mean value of 4.92 in traditional rehabilitation group vs. 4.44 in telerehabilitation group; *p* < 0.01). For this question, the groups differed significantly in the number of people who marked the highest possible answer (“*very satisfied”*). This response was marked by 69% (*n* = 27) and 95% (*n* = 35; *p* = 0.01) in the telerehabilitation and traditional rehabilitation groups, respectively. When asked about their preference for a form of therapy, in the telerehabilitation group, 28% (*n* = 11) of the participants preferred a remote form of treatment, whereas in the traditional rehabilitation group, 5% (*n* = 2) preferred it (*p* < 0.01). In turn, in traditional rehabilitation group, 86% (*n* = 32) of participants selected face-to-face as preferred form of therapy, while in telerehabilitation group, 49% (*n* = 19) preferred this form (*p* < 0.01).

Stepwise multiple regression analysis was used to examine how personality traits could explain the variance in patient satisfaction (Table 3). For the total HCSQ results, agreeableness (*b* = 7.66; CI of 5.01–10.31; *p* < 0.001), conscientiousness (*b* = −3.22; CI of −5.61–−0.84; *p* = 0.01), and extraversion *(b* = −3.22; CI of −6.30–−0.14; *p* = 0.04) were essential predictor variables, accounting for 51% of the variance in patient satisfaction (*p* < 0.001). The same variables were significant predictors of patient satisfaction with the organization of services. Agreeableness (*b* = 9.97; CI of 5.63–14.31; *p* < 0.001), conscientiousness (*b* = −4.96; CI of −8.61–−1.31; *p* = 0.004), and extraversion (*b* = −3.17; CI of −6.01–−0.34; *p* = 0.03) may explain 55% of the variance in satisfaction with the organization of services (*p* < 0.001). Agreeableness (*b* = 13.30; CI of 9.04–17.49; *p* < 0.001), conscientiousness (*b* = −5.19; CI of −8.99–−1.38; *p* = 0.01), and intellect (*b* = −5.75; CI of −10.66–−0.84; *p* = 0.02) were significant predictors of patient satisfaction with service delivery (*p* < 0.001). None of the personality traits showed statistically significant predictive properties for the relationship with the professional.

## 4. Discussion

This study examined the differences in patient satisfaction between telerehabilitation and traditional face-to-face rehabilitation. Based on the HCSQ, the results suggested that patient satisfaction with both forms of rehabilitation was similar. Both groups also did not differ significantly when considering the components of the HCSQ, that is, relationship with the professional, delivery of services, and organization of services. However, considering the results of the single satisfaction question (on a scale of 1–5), those in the telerehabilitation group had a score of 11% lower than those in the traditional rehabilitation group. This difference is most likely due to the number of people who marked the highest possible (“*very satisfied”*) response. The answer was selected by 95% and 69% of the participants in the traditional and telerehabilitation groups, respectively. This difference may be related to the form of therapy itself; however, it may also be due to external factors such as complications with an Internet connection. It is important to note that 13% (*n* = 5) of the study participants reported experiencing issues related to the Internet connection, as indicated by selecting the two lowest possible answers on the survey (“*dissatisfied*” and “*very dissatisfied*”).

The answers to the question regarding the preferred form of therapy are also noteworthy. In the telerehabilitation group, after the therapy, 28% of participants stated that they preferred a remote form of treatment compared to 5% in the traditional rehabilitation group. This difference may be due to participants’ lack of familiarity with the telerehabilitation. Perhaps the mechanism encapsulated in the well-known maxim “*we fear what we do not know*” would explain the difference. Individuals who have not experienced a remote form of therapy may not have positive opinions. On the other hand, those included in the telerehabilitation group were volunteers who wanted to use this form of therapy, which may have skewed their answers to this question because they were more positive toward the telerehabilitation at the beginning.

The results obtained in this study are in line with the other studies on satisfaction with telerehabilitation. In 2017, in a study that used the same satisfaction outcome measure as the one used in this study, Moffet and colleagues [28] concluded that in-home telerehabilitation and standard rehabilitation had a similar level of satisfaction regarding the delivery of healthcare services in patients after total knee arthroplasty. Alsobayel and colleagues [29] conducted a recent preliminary study that found that telerehabilitation had a positive therapeutic impact on pain and function and was deemed an acceptable method of providing physiotherapy services to patients with musculoskeletal conditions. Moreover, in a recent systematic review, Amin and colleagues [30] qualitatively concluded that musculoskeletal telerehabilitation has a similar level of satisfaction compared with a face-to-face consultation among patients and professionals.

Secondarily, this study aimed to determine which personality traits were associated with satisfaction levels. To our knowledge, this is the first study to perform such an analysis regarding telerehabilitation. We hypothesized that agreeableness and extraversion would be predictors of patients’ satisfaction with telerehabilitation. The results of stepwise multiple regression confirmed our hypothesis with the addition of the conscientiousness trait. To summarize the results, higher agreeableness levels and lower conscientiousness and extraversion levels could predict patients’ satisfaction with in-home telerehabilitation. People with high agreeableness adapt more quickly to changing situations [31]. However, they were found to adhere more strictly to the COVID-19 guidelines and engage in more social distancing [32]. Moreover, low agreeableness scores were associated with lower patient satisfaction with healthcare and an increased desire to complain [33]. In turn, social distancing was much more challenging for extroverts, and higher extroversion scores were associated with lower means of social distancing [34,35]. The remote form of therapy makes those with lower levels of extraversion feel more comfortable and derive more satisfaction from the remote support. Notably, none of the personality traits showed significant predictive properties for the relationship with the professional. The lack of variance in results in a particular area may have been due to the consistent application of the same therapeutic approach by a single therapist throughout the study.

In the authors’ opinion, patient satisfaction should be interpreted in terms of clinical effectiveness. Over the past two years, evidence has emerged regarding remote support’s therapeutic efficacy in various rehabilitation fields [14,36]. However, rehabilitation programming is a complex process that requires advanced therapeutic skills and knowledge and taking into account multiple variables and should be aimed at increasing patient adherence to therapy and its clinical effectiveness [37]. In this study, we demonstrated a relationship between personality and patient satisfaction. Previous studies have suggested a possible link between patient satisfaction, higher adherence, and the clinical effectiveness of telerehabilitation [38,39,40]. Personality may indirectly affect clinical effectiveness through the mediating mechanism of patient satisfaction, which should be explored in future research. In this light, personality may be an important consideration when prescribing telerehabilitation programs for patients with different rehabilitation needs.

### Strengths, Limitations, and Future Research Directions

The main strength of this study was its attempt to take a fresh look at the relationship between personality traits and patient satisfaction. While many studies assessing patient satisfaction have used self-developed questionnaires, our study’s use of the standardized HCSQ questionnaire can also be seen as an advantage. The timing of the study during the beginning of the COVID-19 pandemic in Poland and the participants’ lack of previous exposure to remote forms of therapy can be considered both a strength and a limitation. On the one hand, we captured patients’ approach during unprecedented circumstances that had not been studied before. On the other hand, participants’ attitudes towards telerehabilitation may have been influenced by these unique circumstances, potentially limiting the generalizability of our findings.

This study has other limitations that should be taken into consideration. First, the effectiveness of the telerehabilitation intervention was not evaluated. Patient satisfaction should be evaluated in the context of clinical effectiveness, as this is a crucial aspect of therapy. Moreover, the participants in the telerehabilitation group were volunteers, which may have determined their attitude toward this form of treatment at the outset. Additionally, while the aim of this study was to include patients with various musculoskeletal conditions to achieve a wide spectrum of pain origins related to the musculoskeletal system, this approach may be considered a limitation, as it did not focus on specific musculoskeletal conditions. Another potential limitation of this study is its small sample size, which may reduce the generalizability of the findings to a larger population.

Additional research with larger sample sizes and more targeted approaches could provide a deeper understanding of the relationship between personality traits and satisfaction with telerehabilitation for specific musculoskeletal conditions. The above-mentioned relationship should be considered as a secondary outcome measure supplementary to the evaluation of the clinical effectiveness of telerehabilitation. It is possible that specific patient personality traits could influence their satisfaction, which in turn could affect the clinical effectiveness of telerehabilitation. Further investigations are needed to explore potential mediating factors.

## 5. Conclusions

In conclusion, there were no statistically significant differences in patient satisfaction between telerehabilitation and traditional rehabilitation groups. However, subjects participating in telerehabilitation were significantly more likely to indicate this form of therapy as their preferred form, which, in the case of those who had never used remote forms of treatment, could indicate fear of the unknown. In the telerehabilitation group, higher agreeableness levels and lower conscientiousness and extraversion levels could predict patients’ satisfaction with in-home telerehabilitation. In future research, it is recommended to investigate the multidimensional relationship among personality traits, patient satisfaction, and the clinical effectiveness of telerehabilitation. Such research could aid in identifying patient groups that are more likely to benefit from this type of care and inform the development of personalized treatment plans that cater to individual patient needs.

## Figures and Tables

**Table 1 ijerph-20-05019-t001:** Baseline characteristics of the participants.

Variable		Overall	Telerehabilitation	Traditional Rehabilitation	*p*-Value *
*n*		76	39	37	-
*n* (%) of females		45 (59.21)	23 (58.97)	24 (64.86)	0.60
Age, years (*SD*)		35.37 (12.83)	34.41 (12.69)	36.38 (13.07)	0.85
Body mass, kg (*SD*)		68.11 (14.86)	68.03 (13.86)	68.19 (16.04)	0.15
Body height, cm (*SD*)		172.14 (8.98)	172.56 (8.71)	171.70 (9.34)	0.97
VAS, points (*SD*)		7.22 (1.54)	7.18 (1.52)	7.27 (1.58)	0.74
BMI, kg/cm^2^ (*SD*)		22.75 (3.37)	22.63 (2.98)	22.86 (3.77)	0.14
	Normal (BMI 18.5–24.9), *n* (%)	54 (71.05)	28 (71.79)	26 (70.27)	0.89
	Overweight (BMI 25–29.9), *n* (%)	20 (26.32)	11 (28.21)	9 (24.32)	0.70
	Obese (BMI > 30), *n* (%)	2 (2.63)	0 (0.00)	2 (5.40)	0.52
Education, years (*SD*)		16.39 (5.79)	15.72 (3.93)	17.11 (7.25)	0.57
	Elementary/vocational, *n* (%)	2 (2.63)	2 (5.13)	0 (0.00)	0.52
	Secondary, *n* (%)	24 (31.58)	13 (33.33)	11 (29.73)	0.74
	Higher education, *n* (%)	50 (65.79)	24 (61.54)	26 (70.27)	0.42
Pain localization					
	Upper extremity, *n* (%)	25 (32.89)	14 (35.90)	11 (29.73)	0.57
	Lower extremity, *n* (%)	22 (28.95)	11 (28.20)	11 (29.73)	0.88
	Spine/trunk, *n* (%)	29 (38.16)	14 (35.90)	15 (40.54)	0.86

VAS, visual analog scale; BMI, body mass index; SD, standard deviation; * chi-square test or *t*-test, as appropriate.

**Table 2 ijerph-20-05019-t002:** Patient satisfaction with healthcare services.

		Telerehabilitation (*n* = 39)	Traditional Rehabilitation (*n* = 37)	*p*-Value *
HCSQ, % (*SD*)		95.53 (7.28)	95.83 (4.76)	0.33
	Relationship with the professional	98.98 (2.48)	97.97 (3.05)	0.09
	Delivery of services	92.63 (12.15)	92.23 (9.53)	0.31
	Organization of services	94.98 (9.66)	97.30 (3.95)	0.65
Are you satisfied with the therapy form?, mean (*SD*)		4.44 (1.02)	4.92 (0.36)	<0.01
	Very satisfied, *n* (%)	27 (69.23)	35 (94.59)	0.01
	Satisfied, *n* (%)	6 (15.38)	1 (2.70)	0.06
	Moderately satisfied, *n* (%)	3 (7.69)	1 (2.70)	0.33
	Partly satisfied, *n* (%)	2 (5.13)	0 (0.00)	0.59
	Not at all satisfied, *n* (%)	1 (2.56)	0 (0.00)	0.61
Do you prefer either form of therapy?				
	Telerehabilitation, *n* (%)	11 (28.20)	2 (5.40)	<0.01
	Traditional rehabilitation, *n* (%)	19 (48.72)	32 (86.49)	<0.01
	No preference, *n* (%)	9 (23.08)	3 (8.11)	0.07

HCSQ, Health Care Satisfaction Questionnaire; SD, standard deviation; * chi-square test or Mann–Whitney test, as appropriate.

**Table 3 ijerph-20-05019-t003:** Predictors of patient satisfaction within the telerehabilitation group.

		*b* (95% CI)	SE *b*	*t*	*p*-Value	*F*	*p*-Value	R^2^
HCSQ						12.15	<0.001	0.51
	Agreeableness	7.66 (5.01; 10.31)	1.02	5.87	<0.001			
	Conscientiousness	−3.22 (−5.61; −0.84)	−0.41	−2.75	0.01			
	Extraversion	−3.22 (−6.30; −0.14)	−0.31	−2.12	0.04			
Delivery of services						14.40	<0.001	0.55
	Agreeableness	13.30 (9.04; 17.49)	2.08	6.37	<0.001			
	Conscientiousness	−5.19 (−8.99; −1.38)	1.87	−2.77	0.01			
	Intellect	−5.75 (−10.66; −0.84)	2.42	−2.38	0.02			
Organization of services						7.57	<0.001	0.39
	Agreeableness	9.97 (5.63; 14.31)	2.14	4.67	<0.001			
	Conscientiousness	−4.96 (−8.61; −1.31)	1.80	−2.76	0.004			
	Extraversion	−3.17 (−6.01; −0.34)	1.40	−2.27	0.03			

HCSQ, Health Care Satisfaction Questionnaire; CI, confidence interval. Data obtained from the stepwise regression.

## Data Availability

Data are available from the corresponding author on a reasonable request.

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
