# Peer review of "The Impact of Personality Traits on Patient Satisfaction after Telerehabilitation: A Comparative Study of Remote and Face-to-Face Musculoskeletal Rehabilitation during COVID-19 Lockdown"

_ijerph, 2023, doi:10.3390/ijerph20065019_

Round 1

Reviewer 1 Report

The research is interesting because COVID-19 changed the lifestyle of the entire world population. The results were not very significant, which suggests perhaps changing the research paradigm. It would be interesting to carry out the research with a larger number of population with different characteristics than those we have now or to make a comparison in a graph between the results of other authors and their results, which could enrich the discussion and conclusions. Qualitative and quantitative results are needed for the discussion.
Regarding the writing style, the references could be improved, as in line 184 "Moffet et al. concluded...." it would be more appropriate to place the reference after the author's last name.

Reviewer 2 Report

I will now proceed to the revision of the article.

TITLE: The comparison of the 2 methodologies used on patients, i.e. telerehabilitation and face-to-face, should appear, as the whole article talks about this comparison.

INTRODUCTION. Correct.

METHODOLOGY. Important errors;

when a comparison is made, the type of injury of each patient should be discussed (try to focus on a specific pathology, for example, cervicalgia, painful shoulder, dorsalgia, lumbalgia,...).

How has this injury been diagnosed or is it according to the patient's perception (HERE IS THE RISK INVESTIGATOR)?

How the therapies of both groups have been carried out. This is important because depending on the therapy followed, it will probably influence the patients.

RESULTS. Due to the type of methodology the results are very general, not relevant, therefore the conclusion.

IN MY VIEW THE ARTICLE SHOULD BE IMPROVED FOR PUBLICATION and be more specific so that the scientific reader is given better information.

Thank you for reading

Reviewer 3 Report

 I have a few suggestions for improving your article.

- What are the diseases of the participants included in the study? Has this matter been taken into account when dividing into groups?

- Was any randomization created while creating the groups?

- Could the evaluation of both groups through online forms have created a risk of bias in favor of one group?

- I think it would be useful to include a few images in terms of the quality of the work.

Reviewer 4 Report

Dear Authors,

thank you for providing this accurate manuscript. Even though the study was well performed and described, there are some minor remarks i noted:

- There is a mix of british english and american english (e.g., line 31 and 43), this should be formatted uniformly

- Line 167: There is a mix of numbers and words, which is not reader-friendly. It is a nice effort to avoid starting the sentence with a number, but maybe there are other ways that include uniform reporting of numbers as digits or words uniformly.

- Line 187: In running texts the term "and colleagues" blends in better with the manuscripts' language than "et al.". You might consider replacing it.

- Line 192 Shouldn't it read "hypothesizeD" in past term?

- Line 201 Typing error "Extraverts" --> "Extroverts"

Round 2

Reviewer 1 Report

Modifications were made to the document, which allows for a better understanding of the subject, and the references were updated.
The modifications make the article more accessible to the reader.

Reviewer 2 Report

CONGRATULATIONS TO THE AUTHORS FOR THEIR EFFORT AND DEDICATION.

Reviewer 3 Report

In my opinion it may be accepted.